# Replacing Maize Grain with Ancient Wheat Lines By-Products in Organic Laying Hens' Diet Affects Intestinal Morphology and Enzymatic Activity

Nicola Francesco Addeo [1], Basilio Randazzo [2], Ike Olivotto [2], Maria Messina [3], Francesca Tulli [3], Nadia Musco [1,*], Giovanni Piccolo [1], Antonino Nizza [4], Carmelo Di Meo [1] and Fulvia Bovera [1]

1 Department of Veterinary Medicine and Animal Production, University of Napoli Federico II, Via Federico Delpino 1, 80137 Napoli, Italy; nicolafrancesco.addeo@unina.it (N.F.A.); giovanni.piccolo@unina.it (G.P.); carmelo.dimeo@unina.it (C.D.M.); bovera@unina.it (F.B.)
2 Department of Life and Environmental Sciences, Marche Polytechnic University, Via Brecce Bianche, 60131 Ancona, Italy; b.randazzo.live@gmail.com (B.R.); i.olivotto@staff.univpm.it (I.O.)
3 Department of AgriFood, Environment and Animal Science, University of Udine, Via Sondrio, 2, 33100 Udine, Italy; maria.messina@uniud.it (M.M.); francesca.tulli@uniud.it (F.T.)
4 Department of Agricultural Sciences, University of Naples Federico II, 80055 Portici, Italy; nizza@unina.it
* Correspondence: nadia.musco@unina.it

**Abstract:** The effects of replacement of maize grain with ancient wheat by-products on intestinal morphometry and enzymatic activity in laying hens was studied. Eighty hens were divided into two groups (40 each, 8 replicates, 5 hens/replicate) fed two isoproteic and isoenergetic diets. In the treated group, part of the maize was replaced by a mix of ancient grains (AGs) middling, in a 50:50 ratio of *Triticum aestivum* L. var. spelta (spelt) and *Triticum durum dicoccum* L. (emmer wheat). The AG diet affected the weight of all the large intestine tracts, decreasing the weight of caeca ($p < 0.01$) and increasing those of colon ($p < 0.01$), rectum and cloaca ($p < 0.05$). Villus height in the AG group was higher ($p < 0.01$) than the control for the duodenum and jejunum, while for the ileum, the control group showed the highest values ($p < 0.01$). The submucosa thickness was higher ($p < 0.01$) in the control group for the duodenum and ileum, while the jejunum for the AG group showed the highest ($p < 0.05$) submucosa thickness. The crypts depth was higher ($p < 0.01$) in the control group for the duodenum and ileum. Enzyme activity was enhanced by AGs ($p < 0.01$) in the duodenum. Regarding the jejunum, sucrase-isomaltase and alkaline phosphatase had higher activity ($p < 0.05$ and $p < 0.01$, respectively) in the AG group. In the ileum, sucrase-isomaltase showed higher activity ($p < 0.01$) in the control group, while alkaline phosphatase showed the highest values ($p < 0.05$) in the AG group. Overall, results suggested that the dietary inclusion of AGs exerted positive effects in hens, showing an improved intestinal function.

**Keywords:** ancient grains; organic farm; intestinal morphometry; enzymatic activity; animal performance

## 1. Introduction

The term "ancient wheat" indicates primitive grains that never underwent selection or breeding, thus retaining their wild ancestors' patterns, such as high individual variety, brittle rachis, and low harvest index [1]. In the last decade, a rediscovery of ancient varieties took place, aiming to produce high value food products with great health benefits [2,3]. These beneficial properties were attributed to the presence of some nutrients, especially unsaturated fatty acids, soluble fibers, minerals, vitamins, and phytochemicals [3–7]. The highest concentration of such nutrients occurs in the outer layers of grains [8,9], thus explaining why the reduced risk of developing several diseases has been associated with an increased consumption of whole grains [10–12]. After milling, only the endosperm of the whole wheat grain is used to make white flour, whereas the bran and middling are

used as by-products [13], which could also represent an important added value brought to animal feeding. Although durum wheat represents the vast world production of wheat, with the main production and cultivation areas concentrated in the Mediterranean [14], other cultivars such as spelt einkorn (*Triticum monococcum* L.), spelt ("ancient grains") are still produced in small quantities (mainly for traditional foods) in recent years to meet the growing interest of the natural food market.

The compelling argument in favor of ancient wheats is their environmentally friendly production—in fact, growth with few agronomic practices, environmental sustainability, and possible use in marginal lands or under organic growing conditions—thus, they can be produced in a more sustainable way with few external inputs [15,16].

This study is the completion of a previous trial [17] where laying performance, serum biochemistry, and the physical quality of the eggs of hens fed with ancient wheats (spelt and emmer wheat) were investigated. The objective of the present trial, performed in an organic laying hen farm, was to evaluate the effects of a partial replacement of maize grain with local ancient wheats by-products on the intestinal morphometry and brush border enzymatic activity of 36-week-old laying hens.

## 2. Materials and Methods

### 2.1. Animals

The animals used in this study were treated following the Directive 63/2010/EEC regarding animal welfare and the safeguard of experimental animals. This research was approved by the Ethical Animal Care and Use Committee of the University of Naples Federico II, Italy (number 2017/0017676). Experiments were performed in an organic laying hens farm in Avellino (Italy) for 14 weeks (February–May 2019).

Eighty Hy-Line W-36 Single Comb White Leghorn hens, aged 18 weeks and with an average body weight of $1.57 \pm 0.09$ kg, were randomly divided into two groups (40 animals each; 8 replicates of 5 hens each/group). Each group was stabled in a free-range area equipped with an indoor recover for the night. The available space for each hen was 0.45 m$^2$ indoor and 5 m$^2$ in the outdoor areas. A detailed description of the hens' housing system was presented in a previous paper [17].

### 2.2. Diets

Every morning both groups were fed two isoproteic and isoenergetic diets; the differences in ingredients are reported in Table 1.

**Table 1.** Ingredients and chemical composition of the diets used in the trial.

| Ingredients (%) | Control Diet | Ancient Grains Diet |
|---|---|---|
| Maize grain (*Zea mays*) | 59.3 | 25.0 |
| Emmer wheat middling (*Triticum durum dicoccum*) | - | 18.15 |
| Common wheat middling (*Triticum aestivum*) | - | 18.15 |
| Soybean meal | 30.0 | 27.0 |
| Salt | 0.2 | 0.2 |
| Calcium carbonate | 8.0 | 8.0 |
| Vegetable oil | 1.0 | 2.0 |
| Vitamin–mineral premix | 1.0 | 1.0 |
| Monocalcium Phosphate | 0.5 | 0.5 |
| Chemical characteristics | | |
| Dry matter (DM) [1], % | 88.4 | 87.7 |
| Crude protein [1], % DM | 19.5 | 20.2 |
| Ether extract [1], % DM | 4.12 | 4.79 |

**Table 1.** *Cont.*

| Ingredients (%) | | |
|---|---|---|
| | **Control Diet** | **Ancient Grains Diet** |
| Neutral detergent fiber [1], % DM | 10.50 | 13.80 |
| Acid detergent fiber [1], % DM | 5.61 | 5.83 |
| Acid detergent lignin [1], % DM | 0.80 | 0.91 |
| Ca [2], % DM | 3.87 | 3.89 |
| P [2], % DM | 0.52 | 0.54 |
| Methionine [2], % DM | 0.57 | 0.68 |
| Lysine [2], % DM | 1.09 | 1.31 |
| ME [2], Kcal/kg DM | 3.09 | 3.15 |

Vitamin–mineral premix contained the following per kg: retynil acetate 10,000 IU, Vit. D3 3000 IU, Vit. E 45 mg, Vit. B6 4.0 mg, Vit. B12 0.02 mg, Vit. K3 3.5 mg, d-pantothenate calcium 13.9 mg, niacin 50 mg, biotin 0.2 mg, ferrous sulfate 122 mg, cupric sulfate 96 mg, zinc oxide 124 mg, manganese oxide 129 mg, anhydrous calcium iodate 1.5 mg, sodium selenite 0.44 mg. [1] determined values; [2]: calculated values.

The control group (C) received a standard diet containing organic maize and soybean meal and formulated to exceed the requirements reported in the *W-36 Commercial Layers Management Guide* (Hi-line, 2016); in the ancient grains (AG) group around 57.8% of maize grain was substituted with a mix of ancient grains middling in a 50:50 ratio of *Triticum aestivum* L. var. spelta (spelt) and *Triticum durum dicoccum* L. (emmer wheat). The ingredients, ground to the same particle size to avoid a possible influence on the animal feed choice, were mixed in a small local mill; the same mill produced the millings used in the trial. The analysis of the diets (chemical and nutritional) was performed following the AOAC indications [18] (ID number: 2001.12, 978.04, 920.39, 978.10, and 930.05 for dry matter (DM), crude protein (CP), ether extract (EE), crude fiber (CF), and ash, respectively). The methods of Van Soest et al. [19] were used to assay neutral detergent fiber (NDF), acid detergent fiber (ADF), and acid detergent lignin (ADL). The diets' metabolizable energy was calculated according to the NRC [20]. Ca, P, methionine, and lysine contents were estimated according to the diet ingredients content [15,21]. The chemical–nutritional characteristics of the grains are depicted in Table 2.

**Table 2.** Chemical characteristics of the cereal sources used in the trial (% as feed basis).

| Chemical Characteristics | Maize Grain | Spelt Wheat Middlings | Emmer Wheat Middlings |
|---|---|---|---|
| Dry matter [1] % | 88.7 | 87.9 | 89.2 |
| Ash [1] % | 1.3 | 3.4 | 2.8 |
| Crude protein [1] % | 7.65 | 13.2 | 11.8 |
| Ether extract [1] % | 3.7 | 3.5 | 2.7 |
| Neutral detergent fiber [1] % | 8.9 | 21.7 | 13.2 |
| Acid detergent fiber [1] % | 3.2 | 5.4 | 4.3 |
| Acid detergent lignin [1] % | 0.9 | 1.4 | 1.1 |
| Ca [2] % | 0.02 | 0.04 | 0.04 |
| P [2] % | 0.26 | 0.47 | 0.51 |
| ME [2], kcal/kg | 3.32 | 2.87 | 3.43 |

[1] determined values; [2] calculated values.

At the end of the trial (36 weeks of age), after 12 hours of fasting, 16 hens per group (2 per replicate, 32 hens in total) were weighed and slaughtered in a specialized slaughter-house, the digestive tract was removed and weighed, the different intestinal tracts were identified, weighed, measured for length, and properly stored for further analysis.

For each group, small intestine tracts from 8 animals were collected for histological analysis. Samples were washed using a pH 7 isotonic ice-cold saline buffer, dried with absorbent paper, and the duodenum, jejunum, and ileum were separated, weighed, and stored in aluminum foil at −20 °C to be later used for the analysis of the brush border membrane (BBM) enzymes.

### 2.3. Villus and Crypt Morphometry

Samples (0.5 cm) from the duodenum, jejunum, and ileum of 8 animals per group were prepared for histological analysis according to Zarantoniello et al. [22] and Moniello et al. [23]. Briefly, samples were fixed by immersion in 4% phosphate-buffered paraformaldehyde for 48 h. Samples were then washed in phosphate-buffered saline solution (pH = 7.4), dehydrated in graded ethanol solutions, and embedded in paraffin. Cross sections (5 μm) at an interval of 200 μm were stained with Mayer's hematoxylin and eosin (H&E) and Alcian blue (Ab) for the acid mucopolysaccharide-secreting cells (Ab+) detection. Stained sections were examined under a Zeiss Axio Imager A2 microscope according to Zarantoniello et al. [24]. For the evaluation of morphometric parameters (intestinal fold height, submucosa thickness, and crypts depth), 10 random microscopic fields from each section of the duodenum, jejunum, and ileum were acquired by a microscope equipped with a color digital camera Axiocam 503 (Zeiss, Oberkochen, Germany). Data obtained were analyzed by mean of unpaired t test (significance $p < 0.05$) and reported as means $\pm$ SD.

### 2.4. Brush Border Membrane Enzymes Activity

The evaluation of BBM enzymes was done as reported by Shirazy-Beechey et al. [25] with few changes. All steps were done at 4 °C. Briefly, 100 mg of tissue was diluted 1:10 with a buffer (100 mM mannitol, 2 mM Hepes-tris, pH 7.1), with $MgCl_2$ added at a final concentration of 10 mM, and crushed with a TissueLyser (TissueLyser II, Qiagen, Germany) at 30 Hz for 1 min. After a first centrifugation at $2000 \times g$ at 4 °C for 10 min, the supernatant was centrifuged at $15{,}000 \times g$ at 4 °C for 10 min. The resulting supernatant was stored at $-20$ °C until the analysis of maltase, sucrase-isomaltase (SI), L-aminopeptidase (L-ANP), and alkaline phosphatase (IAP) BBM enzyme activity.

The hydrolysis of sucrose and maltose by the mucosal maltase and sucrase was determined as reported by Tibaldi et al. [26].

The intestine alkaline phosphatase (IAP) activity was determined by using a commercial kit (Paramedical, Pontecagnano Faiano, Sa, Italy) using the manufacturer's instructions.

L-aminopeptidase (L-ANP) was determined as reported by Vizcaino et al. [27].

Total proteins were determined according to Bradford [28] (Sigma-Aldrich cat. no. B6916) and bovine serum albumin (Sigma-Aldrich cat. no. 0834) as a standard.

One unit (U) of enzyme activity is the amount of enzyme that transforms or hydrolyzes 1 μmol of the substrate per minute. Specific enzyme activity was calculated as 1 U of the enzyme activity per mg of protein.

### 2.5. Statistical Analysis

The normal distribution of data and error was evaluated using the Shapiro–Wilk test (SAS, 2002). Data were processed by one-way ANOVA according to the following model:

$$Y_{ij} = m + D_i + e_{ij}$$

where Y is the single observation, m the general mean, D the effect of the diet (i = control or ancient grains), e is the error using the PROC GLM [20]. The comparison between the means was performed by Tukey's test [29]. The results were expressed as average value and the significance level was set at $p \leq 0.05$; $p$ values $< 0.10$ were considered as a tendency.

## 3. Results

The relative weights of the whole digestive tract, proventriculus, gizzard, liver, spleen, and abdominal fat for the different dietary treatments are reported in Table 3.

**Table 3.** Body weight and relative weight of empty gut, proventriculus, gizzard, liver, spleen, and abdominal fat in hens fed the experimental diets over 16 weeks.

|  | Control Diet | Ancient Grains Diet | RMSE | *p*-Value |
|---|---|---|---|---|
| Body weight, g | 1498.7 | 1521.2 | 97.2 | 0.6441 |
| Empty gut, % BW | 8.02 | 8.84 | 0.71 | 0.0725 |
| Gizzard, % BW | 1.40 | 1.50 | 0.27 | 0.1106 |
| Proventriculus, % BW | 0.51 | 0.53 | 0.07 | 0.0854 |
| Liver, % BW | 2.81 | 2.65 | 0.21 | 0.4396 |
| Spleen, % BW | 0.14b | 0.19a | 0.05 | 0.0316 |
| Abdominal fat, % BW | 1.76a | 0.63b | 0.086 | 0.0112 |

RMSE: root mean square error; a, b: $p < 0.05$. $n = 16$.

The spleen weight expressed as BW percentage was higher in the AG group ($p < 0.05$), while the percentage of abdominal fat showed higher values in the control group ($p < 0.05$).

Table 4 shows the relative weights and lengths of the small intestine tracts of the hens, according to the dietary treatments. The duodenum and jejunum lengths exhibited the highest weights in the control group ($p < 0.05$).

**Table 4.** Relative weight and length (% body weight) of the empty duodenum, jejunum, and ileum in hens fed the experimental diets over 16 weeks.

|  | Control Diet | Ancient Grains Diet | RMSE | *p*-Value |
|---|---|---|---|---|
| Duodenum weight | 1.40 | 1.41 | 0.11 | 0.8439 |
| Jejunum weight | 1.73 | 1.70 | 0.83 | 0.6241 |
| Ileum weight | 1.03 | 1.15 | 0.09 | 0.4744 |
| Duodenum length | 2.52a | 2.16b | 0.19 | 0.0156 |
| Jejunum length | 4.70a | 4.12b | 0.36 | 0.0372 |
| Ileum length | 3.40 | 3.28 | 0.35 | 0.6636 |

RMSE: root mean square error; a, b: $p < 0.05$. $n = 16$.

Table 5 reports the relative weights and lengths of the large intestine tracts of the hens according to the dietary treatments. Ancient grains affected the weight of all the large intestine tracts, decreasing the relative weight of caeca ($p < 0.01$) and increasing that of colon ($p < 0.01$), rectum and cloaca ($p < 0.05$). Conversely, only the relative length of the caeca was significantly decreased by the treatment ($p < 0.05$). Colon length % tended ($p = 0.0510$) to be higher in the AG group.

**Table 5.** Relative weight and length (% body weight) of the empty caeca, colon, rectum, and cloaca in hens fed the experimental diets over 16 weeks.

|  | Control Diet | Ancient Grains Diet | RMSE | *p*-Value |
|---|---|---|---|---|
| Caeca weight | 0.66A | 0.61B | 0.06 | 0.0002 |
| Colon weight | 0.49B | 0.63A | 0.05 | 0.0126 |
| Rectum weight | 0.26b | 0.31a | 0.02 | 0.0262 |
| Cloaca weight | 0.56b | 0.64a | 0.04 | 0.0356 |
| Caeca length | 2.75a | 2.42b | 0.24 | 0.0371 |
| Colon length | 1.21 | 1.40 | 0.10 | 0.0510 |
| Rectum length | 0.64 | 0.62 | 0.04 | 0.1722 |
| Cloaca length | 0.27 | 0.28 | 0.05 | 0.1798 |

RMSE: root mean square error; A, B: $p < 0.01$; a, b: $p < 0.05$. $n = 16$.

Table 6 shows the morphometric evaluation performed on the duodena, jejuna, and ilea of hens according to the dietary treatment. The villus length in the AG group was higher ($p < 0.01$) than the control in the duodenum and jejunum while in the ileum, the control group showed the highest values of villus length ($p < 0.01$).

**Table 6.** Morphometric evaluations performed on the duodena, jejuna, and ilea of hens fed the experimental diets over 16 weeks.

|  | Control Diet | Ancient Grains Diet | RMSE | *p*-Value |
|---|---|---|---|---|
|  | Villus length, μm | | | |
| Duodenum | 888.1B | 1116.5A | 103.5 | <0.0001 |
| Jejunum | 941.8B | 1156.8A | 108.5 | <0.0001 |
| Ileum | 953.2A | 660.6B | 198.9 | 0.0016 |
|  | Submucosa thickness, μm | | | |
| Duodenum | 64.2A | 33.6B | 6.63 | 0.0002 |
| Jejunum | 43.22b | 51.65a | 7.72 | 0.0138 |
| Ileum | 71.28A | 45.39B | 18.98 | 0.0030 |
|  | Crypts depth, μm | | | |
| Duodenum | 410.58A | 317.0B | 45.590 | <0.0001 |
| Jejunum | 434.9 | 360.1 | 115.7 | 0.1274 |
| Ileum | 419.4A | 263.2B | 10.41 | 0.0010 |
|  | Villus:Crypt ratio | | | |
| Duodenum | 2.14B | 3.52A | 0.18 | 0.0043 |
| Jejunum | 2.16B | 3.21A | 0.20 | 0.0062 |
| Ileum | 2.27b | 2.50a | 0.16 | 0.0121 |

RMSE: root mean square error; A, B: $p < 0.01$; a, b: $p < 0.05$. $n = 8$.

The submucosa thickness was higher ($p < 0.01$) in the control group for the duodenum and ileum, while the jejunum in the AG group showed the highest ($p < 0.05$) submucosa thickness. The crypts depth was higher ($p < 0.01$) in the control group for the duodenum and ileum. The villus:crypt ratio in the three tracts of the small intestine was higher for the AG group in the duodenum and jejunum ($p < 0.01$) and in the ileum tract ($p < 0.05$).

Table 7 shows the specific activity of the brush border enzymes in the small intestine of hens according to the dietary treatment. The activity of all the evaluated enzymes (maltase, sucrase-isomaltase, L-aminopeptidase, and intestinal alkaline phosphatase) was enhanced by the presence of AGs ($p < 0.01$) in the duodenum.

**Table 7.** Specific activity of intestinal brush border membrane (BBM) enzymes measured in the different digestive tracts of the hens fed the experimental diets over 16 weeks.

| Enzymatic Activity | Control Diet | Ancient Grains Diet | RMSE | *p*-Value |
|---|---|---|---|---|
| *Duodenum* | | | | |
| Maltase, U | 22.49B | 46.94A | 7.82 | 0.000 |
| SI, U | 5.95B | 13.10A | 4.29 | 0.005 |
| L-ANP, U | 1.58B | 3.28A | 0.67 | 0.000 |
| IAP, mU | 290.83B | 550.61A | 131.58 | 0.001 |
| *Jejunum* | | | | |
| Maltase, U | 32.97 | 41.19 | 17.16 | 0.339 |
| SI, U | 7.43b | 13.35a | 4.81 | 0.023 |
| L-ANP, U | 2.44 | 2.25 | 0.83 | 0.656 |
| IAP, mU | 397.33B | 817.57A | 245.20 | 0.003 |
| *Ileum* | | | | |
| Maltase, U | 34.18 | 32.23 | 8.84 | 0.651 |
| SI, U | 10.77A | 6.26B | 2.29 | 0.002 |
| L-ANP, U | 2.73 | 2.48 | 0.544 | 0.360 |
| IAP, mU | 321.16b | 493.26a | 138.18 | 0.032 |

SI: sucrase-isomaltase; L-ANP: L-aminopeptidase; IAP: intestinal alkaline phosphatase. RMSE: root mean square error; A, B: $p < 0.01$; a, b: $p < 0.05$. $n = 8$.

Regarding the jejunum, SI and IAP had higher activity ($p < 0.05$ and $p < 0.01$, respectively) in the AG group than in the control group. In the ileum, SI showed higher activity ($p < 0.01$) in the control group, while IAP showed the highest values ($p < 0.05$) in the AG group.

Histological analysis performed on the duodenum, jejunum, and ileum exhibited intact intestinal mucosa in both groups in all the analyzed samples, showing a continuous epithelial layer forming the absorptive mucosa and a low number of exfoliated cells in the lumen (Figure 1).

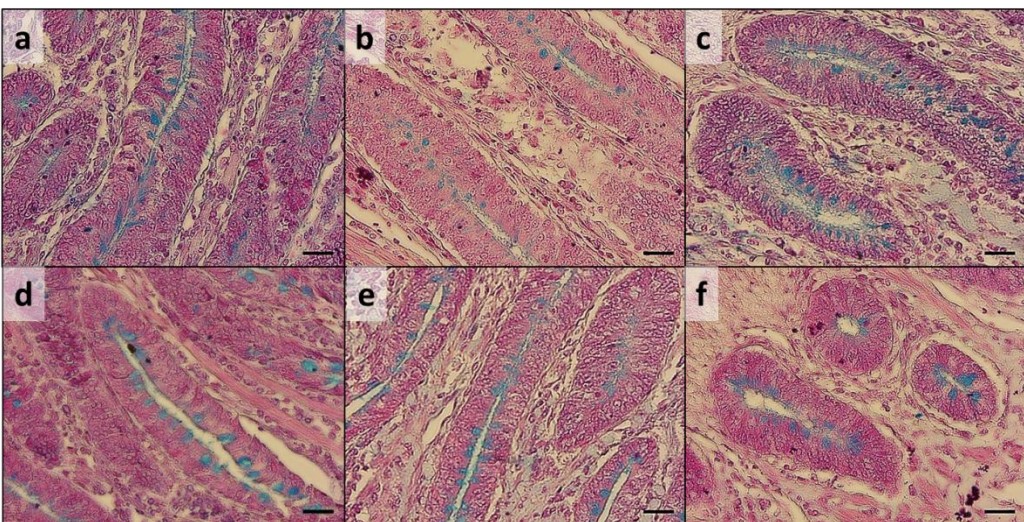

**Figure 1.** Ab+ mucous cells (arrowheads) in duodenum (**a**,**d**), jejunum (**b**,**e**), and ileum (**c**,**f**) crypts from C (**a**–**c**) and AG groups (**d**–**f**) of hens fed the experimental diets over 16 weeks. Alcian blue. Scale = 20 μm.

A regular degree of basal crypts was observed in all the intestinal traits analyzed, with a comparable Ab+ mucous cells distribution between the experimental groups (Figure 2). Nevertheless, melanomacrophage intra-epithelial influx (Figure 2) was observed in 75% of group C duodenum samples, while it was not detected in any of the AG group samples.

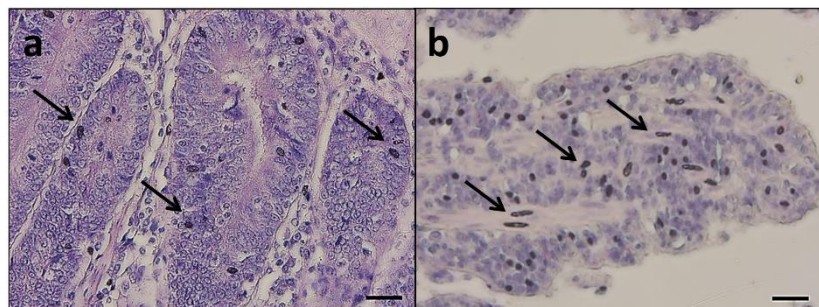

**Figure 2.** Group C duodenum crypts (**a**) and lamina propria of jejunum villi (**b**) showing melanomacrophage influx (arrows) in hens fed the experimental diets over 16 weeks. H&E. Scale = 20 μm.

## 4. Discussion

The inclusion of ancient grains in the diet of hens did not affect the weight of the small intestine tracts mainly involved in the digestion and absorption of the nutrients, but the length of both duodenum and jejunum was lower by 13.36 and 11.10% than the control group, respectively. According to the first part of this trial [17], feed intake was lower in the hens of the AG group than in the control one (120.9 vs. 134.1 g/d, respectively, $p < 0.01$), probably due to a low palatability of the ancient wheats compared to the maize [17], while the digestibility of organic matter (75.12 vs. 75.98%), crude protein (86.23 vs. 85.47%) and ether extract (92.48 vs. 91.87%) were unaffected by the dietary treatment. The higher levels in methionine and lysine are due to their higher content in ancient wheats than in maize.

However, the essential amino acid levels were calculated based on literature values and not determined.

The higher feed intake could justify the higher percentage of abdominal fat recorded in hens fed the control diet, in contrast with the results of El-Katcha et al. [30]. These authors reported that including wheat (in ratios of 25 and 50%) instead of corn in the diet of broiler chickens did not significantly reduce abdominal fat weight compared with a broiler chicken group fed a corn–soybean-based diet. However, the previous trial from Lombardi et al. [17] showed that wheat inclusion did not impair intestinal nutrient digestibility and the hen's final weight.

Such a result could be related to the different intestinal morphology between the groups. The morphology of the small intestine is often used to explore its functionality and, generally an increased villus height, as recorded in the present trial in the duodenum and jejunum for the AG group, is indicative of an improved intestinal function [31]. It seems that when ancient grains were administered in the diet, the hens' intestinal tracts increased absorptive tissue rather than length, which would involve a higher energy requirement. The improvement in intestine absorptive surface may be related to the presence of AG peptides and to the presence of insoluble non-polysaccharide starch (NPS) which, different from soluble viscous NSP, is able to enhance intestine welfare [32,33]. In any case, the effect of bioactive peptides from cereal grains has already been recognized in humans [34,35]. Another interesting consideration is that, in chickens, the ileal villi are smaller than those of the previous tracts of the small intestine, and, in hens fed corn–soybean-based diets, very few nutrients are available beyond the jejunum [36–38].

In addition, in the present trial, the highest feed intake recorded in the control group, associated with an unmodified nutrient digestibility in comparison to the AG group [17], could have produced a higher amount of undigested starch particles in the ilea of the hens fed the corn–soybean-based diet, and this could be responsible for the higher height of villi in the ilea of hens of the control group. It is noteworthy that the ileum plays a significant role in digestion and absorption of undigested starch in chickens [39,40].

Also, Yamauchi [41] reported that increasing the load of nutrients from the duodenum to the ileum (for both jejunum dissection and different diets) may stimulate the absorptions in the ileum, thus resulting in a compensatory development of the villi as observed in the control group ilea. It is known that the presence of longer villi is due to the activation of cell mitosis in the crypts [42]; as a consequence, a larger crypt area means an increased cell production. In our trial, only the crypt depth was recorded, being higher in the control group for both duodenum and ileum tracts. In addition, the villus height-to-crypt depth ratio (VH/CD), which is a measure of the epithelial cell turnover [41], was lower in the control group for the three small intestine tracts. Sozcu and Ipek [43] studied the effects of lignocellulose supplementation on jejunal histomorphology of laying hens, including increasing levels (0.05, 0.1, and 0.2 g/100 g feed) of a commercial product containing 92.6 g of ADL per 100 g. The authors observed that lignocellulose at 0.05 and 0.1% improved jejunum mucosal development by increasing villus height and VH/CD ratio, and such a result is in agreement with our findings, since the ADL in the AG group was 0.14% higher than the control. The villus size (height and width) is fundamental for the absorptive activity of the intestine [44]. As a result, a higher VH and CD ratio can be recognized as marker for an increase in the digestion and absorption of nutrients [43]. Moreover, dietary AGs in addition to modifying villus size enhanced the brush border membrane enzymatic activities, contributing to normalizing the nutrient digestion.

The levels of brush border membrane enzymes could have also contributed to normalizing the nutrient digestion despite the lower intestinal length and feed intake of hens fed AGs. The digestion and absorption of almost all the nutrients in the diet take place in the small intestine [45]: around 95% of the fats are digested in the duodenum [46]; fats, starch, and protein are digested in the jejunum and ileum [47], but the ileum is mainly involved in water and mineral absorption [45]. Disaccharidase and peptidase are extrinsic enzymes, which mainly participate in the digestion and absorption of nutrients. An increase in the

activities of disaccharidase and peptidase is connected with enhanced nutrient digestion and absorption capacity of the intestinal epithelium [48]. In the duodenum, all the BBM enzyme activities in the AG-fed hens were higher than the control: +109% for maltase, +120 for SI, +108 for L-ANP, and +89.3% for IAP. In the jejunum, only SI (+79.68%) and IAP (+105.8%) were higher in the AG group, while in the ileum, SI showed a lower activity (−72.04%) and IAP a higher activity (+53.6%) in the AG group. Such an increase of the BBM enzyme activity in the hens fed the AGs diet is coherent with the histological observations in the duodenum, where an increase in villus height and villus:crypt ratio was observed. The increased enzyme activity could be related both to a higher enterocyte number and to the quantity and composition of the digesta since malt, SI, and L-ANP are substrate-inducible enzymes. Other features together with the BBM enzyme activity could have affected the absorption of nutrients in hens fed the AGs. Among these, PepT-1 and Na+/K+ ATPase expression and activity, and Na+ availability to co-transport, may have played a role in the substantially equal ileal nutrient digestibility between the experimental diets. Also in Tibaldi et al. [26] the differences related to the BBM enzyme activities were not consistent with the changes in nutrient digestibility in European sea bass.

The different activities of intestinal enzymes are affected, among the other factors, by the diet [49,50]. As the main difference in the two groups was the carbohydrate source, this aspect must be focalized. Even if not measured in the present trial, it is well known that corn is richer in starch, while wheats have higher non-starch polysaccharides (NSPs) contents, which act as anti-nutritional factors in poultry [51]. Accordingly, the AG diet contained 6.98% of hemicelluloses (NDF–ADF) [19] and the control diet only 4.32%. According to De Keyser et al. [52], corn contains 8.74% DM of NSP, and wheat 9.93%. Gebruers et al. [53] stated that emmer contained about half the level of mixed-linkage beta-glucan (0.25–0.45% of DM) present in winter, spring, and spelt wheats (0.50–0.95% of DM). Being indigestible by poultry enzymes, the dietary NSPs undergo a microbial digestion that increases along the gastrointestinal tract, including the upper small intestine [54–57]. The consequent fermentation of NSPs leads to the production of volatile fatty acids and is able to decrease the load of carbohydrate necessary for the development of intestinal enzyme function [58]. However, a decrease of intestinal enzyme function due to the presence of soluble NSPs can be compensated by cellular hyperplasia and hypertrophy [59]. Some studies on commercial NSPs [60] reported higher activities of maltase, sucrase, and alkaline phosphatase in the jejuna of broiler chickens fed diets including xanthan gum. In the ileum, maltase activity was also found marginally increased, thus suggesting that NSPs, other fiber sources, or their metabolites should be involved in the maturation of gut cells, which is crucial for various functions along the villus:crypt axis [58].

The cereal type affected jejunal mucosal disaccharidases. In fact, the increase in intestinal enzymes is mechanically improved when the chyme passes through the digestive tract [61]. As reported by Shakouri et al. [62], the bulk of digesta in the gut of birds fed maize diets, resulting from a higher feed intake than birds fed on barley and wheat, may increase maltase activity.

The increased disaccharidase activity facilitates starch digestion, thus increasing the amount of energy absorbable from feeds and giving to the animals a metabolic advantage, despite the lower amount of dietary starch. Our results indicated that the jejunum and ileum are less sensitive to the effect of the diet, probably because the feeds get there partially digested by the previous intestinal tract. In the jejunum and ileum, the maltase and L-ANP showed similar activity in both the experimental groups. In general, the SI activity was higher in the jejuna and lower in the ilea of the AG group. Our results partially agree with Kohl et al. [50], who observed an effect of dietary starch on maltase activity in the mid intestine of chickens.

The aminopeptidases located in the intestinal brush border, enzymes known to carry out intestinal membrane digestion [27], were significantly affected by AGs inclusion in the hen's diet, while IAP has been traditionally considered a marker of the enterocyte maturation and is involved in the dephosphorylation of the microbial LPS, thus preventing

its toxicity [63,64]. These enzymes, in combination with other intestinal membrane-bound enzymes, are important for the absorption of the nutrients in order to keep homeostasis [65]. The increased activity of these intestinal enzymes in the AG-fed hens was consistent with the villi height in the duodenum and jejunum and with the villus:crypt ratio in all the considered intestinal tracts. In previous studies, a modulatory effect of the dietary starch and protein level on the aminopeptidases and disaccharidases activity was not observed in chickens [50]. Therefore, we hypothesize a stimulatory effect of the specific AG composition on the metabolism and development of the enterocytes together with a positive effect on the activity of the IAP and L-ANP.

Furthermore, a high level of IAP activity provides the animals a better defense [63,64].

An additional role of IAP is the dephosphorylation of dietary proteins/peptides, so its high activity in the AG group could be also related to a particular moiety of proteins from the digested AGs.

## 5. Conclusions

To summarize, based on the histological and physiological analysis of the gut performed in the present study, we suggest that the dietary inclusion of ancient grains is related to positive effects in hens. In particular, the increase of both the digestive enzymes activity and the villus height are indicative of an improved intestinal function without affecting growth and nutrient digestibility, as shown in a previous study [17].

In conclusion, these data characterize the opportunity to include ancient grains in the hen's diet to noticeably affect the structure as well as the overall digestive enzyme activity.

**Author Contributions:** Conceptualization, F.B. and I.O.; methodology, B.R., M.M. and F.T.; software, F.B.; validation, F.B., M.M., F.T., G.P., A.N.; formal analysis, N.F.A., B.R., M.M.; investigation, N.F.A., N.M., G.P.; data curation, N.F.A., N.M., C.D.M. and F.T.; writing—original draft preparation, F.B., C.D.M., G.P., I.O. and F.T.; writing—review and editing, N.F.A. and N.M. All authors have read and agreed to the published version of the manuscript.

**Funding:** This research received no external funding.

**Institutional Review Board Statement:** This research was approved by the Ethical Animal Care and Use Committee of the University of Naples Federico II, Italy (number 2017/0017676). Experiments were per-formed in an organic laying hens farm in Avellino (Italy) for 14 weeks (February–May 2019).

**Informed Consent Statement:** Not applicable.

**Data Availability Statement:** Data are available from the authors upon reasonable request.

**Conflicts of Interest:** The authors declare no conflict of interest.

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
