# Peer review of "Replacing Maize Grain with Ancient Wheat Lines By-Products in Organic Laying Hens’ Diet Affects Intestinal Morphology and Enzymatic Activity"

_sustainability, doi:10.3390/su13126554_

Round 1
Reviewer 1 Report
The paper is interesting but it is lacking the performance results which were published elsewhere.
In the introduction, I would like to see more publications of ancient wheat lines. In the material and methods, how were the middlings produced?
In the dicussion, the results show higher villi and more enzyme activity in the upper small intestine with ancient wheat lines. I would like to see more explanations to this and why was digestibility the same.
Some comments:
L89. Is 57,8% the right figure?
Table 2. Is it right that ME value of emmer wheat middling is higher than that of maize?
L146 Was there normality check of the data? If not, they should do it.
L229 Lower ME content of corn diets increased the feed intake of it.
L241-242 Explain more about this effect.
L338-339 You cannot say anything about the performance because it was reported elsewhere.
Author Response
Dear reviewer,
thank you for your valuable comments. Please find attached the responses to your queries.
Kind regards,
Nadia Musco

Reviewer 2 Report
see attachment.

Author Response

(The authors gave the same response as above.)
